# Experimental Design for Cost-Aware Learning of Causal Graphs

**Erik M. Lindgren**
University of Texas at Austin
erikml@utexas.edu

**Murat Kocaoglu**
MIT-IBM Watson AI Lab
murat@ibm.com

**Alexandros G. Dimakis**
University of Texas at Austin
dimakis@austin.utexas.edu

**Sriram Vishwanath**
University of Texas at Austin
sriram@ece.utexas.edu

## Abstract

We consider the minimum cost intervention design problem: Given the essential graph of a causal graph and a cost to intervene on a variable, identify the set of interventions with minimum total cost that can learn any causal graph with the given essential graph. We first show that this problem is NP-hard. We then prove that we can achieve a constant factor approximation to this problem with a greedy algorithm. We then constrain the sparsity of each intervention. We develop an algorithm that returns an intervention design that is nearly optimal in terms of size for sparse graphs with sparse interventions and we discuss how to use it when there are costs on the vertices.

## 1 Introduction

Causality is a fundamental concept in science and an essential tool for multiple disciplines such as engineering, medical research, and economics [28, 27, 29]. Discovering causal relations has been studied extensively under different frameworks and under various assumptions [25, 15]. To learn the cause-effect relations between variables without any assumptions other than basic modeling assumptions, it is essential to perform experiments. Experimental data combined with observational data has been successfully used for recovering causal relationships in different domains [30].

There is significant cost and time required to set up experiments. Often there are many ways to design experiments to discover cause-and-effect relationships. Considering cost when designing experiments can critically change the total cost needed to learn the same causal system. King et al. [18] created a robot scientist that would automatically perform experiments to learn how a yeast gene functions. Different experiments required different materials with large variations with costs. By considering material cost when defining interventions, their robot scientist was able to learn the same causal structure significantly cheaper.

Since the work of King et al., there have been a number of papers on automated and cost-sensitive experiment design for causal learning in biological systems. Sverchkov and Craven [33] discuss some aspects on how to design costs. Ness et al. [24] develop an active learning strategy for cost-aware experiments in protein networks.

We study the problem of cost-aware causal learning in Pearl's framework of causality [25] under the causal sufficiency assumption, i.e., when there are no latent confounders. In this framework, there is a directed acyclic graph (DAG) called the *causal graph* that describes the causal relationships between the variables in our system. Learning direct causal relations between the variables in the system is equivalent to learning the directed edges of this graph. From observational data, we can learn of the existence of a causal edge, as well as some of the edge directions, however in general we cannot learn

the direction of every edge. To learn the remaining causal edges, we need to perform experiments and collect additional data from these experiments [3, 10, 13].

An intervention is an experiment where we force a variable to take a particular value. An intervention is called a stochastic intervention when the value of the intervened variable is assigned to another independent random variable. Interventions can be performed on a single variable, or a subset of variables simultaneously. In the non-adaptive setting, which is what we consider here, all interventions are performed in parallel. In this setting, we can only guarantee that an edge direction is learned when there is an intervention such that exactly one of the endpoints is included [19].

In the minimum cost intervention design problem, as first formalized by Kocaoglu et al. [19], there is a cost to intervene on each variable. We want to learn the causal direction of every edge in the graph with minimum total cost. This becomes a combinatorial optimization problem, and so two natural questions that have not yet been addressed are if the problem is NP-hard and if the greedy algorithm proposed by [19] has any approximation guarantees.

Our contributions:

- We show that the minimum cost intervention design problem is NP-hard.
- We modify the greedy coloring algorithm proposed in [19]. We establish that our modified algorithm is a $(2 + \varepsilon)$-approximation algorithm for the minimum cost intervention design problem. Our proof makes use of a connection to submodular optimization.
- We consider the sparse intervention setup where each experiment can include at most $k$ variables. We show a lower bound to the minimum number of interventions and create an algorithm which is a $(1 + o(1))$-approximation to this problem for sparse graphs with sparse interventions.
- We introduce the minimum cost $k$-sparse intervention design problem and develop an algorithm that is essentially optimal for the unweighted variant of this problem on sparse graphs. We then discuss how to extend this algorithm to the weighted problem.

## 2 Minimum Cost Intervention Design

### 2.1 Relevant Graph Theory Concepts

We first discuss some graph theory concepts that we utilize in this work.

A proper *coloring* of a graph $G = (V, E)$ is an assignment of colors $c : V \mapsto \{1, 2, \ldots, t\}$ to the vertices $V$ such that for all edges $uv \in E$ we have $c(u) \neq c(v)$. The *chromatic number* is the minimum number of colors needed for a proper coloring to exist and is denoted by $\chi$.

An *independent set* of a graph $G = (V, E)$ is a subset of the vertices $S \subseteq V$ such that for all pairs of vertices $u, v \in S$ we have that $uv \notin E$. The independence number is the size of the maximum independent set and is denoted by $\alpha$. If there is a weight function on the vertices, a maximum weight independent set is an independent set with the largest total weight.

A *vertex cover* of a graph $G = (V, E)$ is a subset of vertices $S$ such that for every edge $uv \in E$, at least one of $u$ or $v$ are in $S$. Vertex covers are closely related to independent sets: if $S$ is a vertex cover then $V \setminus S$ is an independent set and vice versa. Further, if $S$ is a minimum weight vertex cover then $V \setminus S$ is a maximum weight independent set. The size of the smallest vertex cover of $G$ is denoted $\tau$.

A *chordal graph* is a graph such that for any cycle $v_1, v_2, \ldots, v_t$ for $t \geq 4$, there is a chord, which is an edge between two vertices that are not adjacent in the cycle. There are linear complexity algorithms for finding a minimum coloring, maximum weight independent set, and minimum weight vertex cover of a chordal graph. Any induced subgraph of a chordal graph is also a chordal graph.

Given a graph $G = (V, E)$ and a subset of vertices $I \subseteq V$, the *cut* $\delta(I)$ is the set of edges $uv \in E$ such that $u \in I$ and $v \in V \setminus I$.

### 2.2 Causal Graphs and Interventional Learning

Consider two variables $X, Y$ of a system. If every time we change the value of $X$, the value of $Y$ changes but not vice versa, then we suspect that variable $X$ *causes* $Y$. If we have a set of variables, the

same intuition carries through while defining causality. This asymmetry in the directional influence between variables is at the core of causality.

Pearl [25] and Spirtes et al. [32] formalized the notion of causality using directed acyclic graphs (DAGs). DAGs are suitable to encode asymmetric relations. Consider a system of $n$ random variables $\mathcal{V} = \{V_1, V_2, \ldots, V_n\}$. The structural causal model of Pearl models the causal relations between variables as follows: each variables $V_i$ can be written as a deterministic function of a set of other variables $S_i$ and an unobserved variable $E_i$ as $V_i = f_i(S_i, E_i)$. We assume that $E_i$, called an exogenous random variable, is independent from everything, i.e., every variable in $\mathcal{V}$ and all other exogenous variables $E_j$. The graph that captures these directional relations is called the causal graph between variables in $\mathcal{V}$. We restrict the graph created to be acyclic, so that if we replace the value of a variable we potentially change the descendent variables but the ancestor variables will not change.

Given a causal graph, a variable is said to be caused by the set of parents [1]. This is precisely $S_i$ in the structural causal model. It is known that the joint distribution induced on $\mathcal{V}$ by a structural causal model factorizes with respect to the causal graph. Thus, the causal graph $D$ is a valid Bayesian network for the observed joint distribution.

There are two main approaches for learning causal graphs from observational distribution: *i) score based* [6, 11], and *ii) constraint based* [25, 32]. Score based approaches optimize a score (e.g., likelihood) over all Bayesian networks to recover the most likely graph. Constraint-based approaches, such as IC and PC algorithms, use conditional independence tests to identify the causal edges that are invariant across every graph consistent with the observed data. This remaining mixed graph is called the *essential graph*. The undirected components of the essential graph are always *chordal* [32, 9]

Although PC runs in time exponential in the maximum degree of the graph, various extensions make it feasible to run it even on graphs with 30,000 nodes with maximum degree up to 12 [26]. To learn the rest of the causal edge directions without additional assumptions, we need to use *interventions* on the undirected, chordal components. [2] An intervention is an experiment where a random variable is *forced* to take a certain value. Due to the acyclicity assumption on the graph, if $X \to Y$, then intervening on $Y$ should not change the distribution of $X$, however intervening on $X$ will change the distribution of $Y$. Running the observational learning algorithms like PC/IC after an intervention on a set $S$ of variables, we can learn the new skeleton after the intervention, which allows us to identify the immediate children and immediate parents of the intervened variables. Therefore, if we perform a randomized experiment on a set $S$ of vertices in the causal graph, we can learn the direction of all the edges cut between $S$ and $V \setminus S$. This approach has been heavily used in the literature [13, 10, 31].

## 2.3 Graph Separating Systems and Minimum Cost Intervention Design

Given a causal DAG $D = (V, E)$, we observe the essential graph $\mathcal{E}(D)$. Kocaoglu et al. [19] established that if we want to guarantee learning the direction of the undirected edges with nonadaptive interventions, it is nessesary and sufficient for our intervention design $\mathcal{I} = \{I_1, I_2, \ldots, I_m\}$ to be a *graph separating system* on the undirected component of the graph $G$.

**Definition 1** (Graph Separating System). Given an undirected graph $G = (V, E)$, a *graph separating system* of size $m$ is a collection of $m$ subsets of vertices $\mathcal{I} = \{I_1, I_2, \ldots, I_m\}$ such that every edge is cut at least once, that is, $\bigcup_{I \in \mathcal{I}} \delta(I) = E$.

Recall that the undirected component of the essential graph of a causal DAG is always a chordal graph. We can now define the minimum cost intervention design problem.

**Definition 2** (Minimum Cost Intervention Design). Given a chordal graph $G = (V, E)$, a set of weights $w_v$ for all $v \in V$, and a size constraint $m \geq \lceil \log \chi \rceil$, the *minimum cost intervention design problem* is to find a graph separating system $\mathcal{I}$ of size at most $m$ that minimizes the cost

$$\text{cost}(\mathcal{I}) = \sum_{I \in \mathcal{I}} \sum_{v \in I} w_v.$$

Graph separating systems are tightly related to graph colorings. Mao-Cheng [22] proved that the smallest graph separating system has size $m = \lceil \log \chi \rceil$, where $\chi$ is the chromatic number. To see this, for each vertex, we create a binary vector $c(v)$ where $c(v)_i = 1$ if $v \in I_i$ and $c(v)_i = 0$ if $v \notin I_i$. Since two neighboring vectors $u$ and $v$ must have, for some intervention $I_i$, exactly one of $u \in I_i$ or $v \in I_i$, the assignment of vectors to vertices $c : V \mapsto 0, 1^m$ is a proper coloring. With a size $m$ graph separating system, we are able to create $2^m$ different colors, proving that the size of the smallest separating system is exactly $m = \lceil \log \chi \rceil$.

The equivalence between graph separating systems and coloring allows us to define an equivalent coloring version of the minimum cost intervention design problem, which was first developed in [19].

**Definition 3** (Minimum Cost Intervention Design, Coloring Version). Given a chordal graph $G = (V, E)$, a set of weights $w_v$ for all $v \in V$, and the colors $C = \{0, 1\}^m$ such that $|C| \geq \chi$, the *coloring version of the minimum cost intervention design problem* is to find a proper coloring $c : V \mapsto C$ that minimizes the total cost

$$\text{cost(c)} = \sum_{v \in V} \|c(v)\|_1 w_v.$$

Given a minimum cost coloring from the coloring variant of the minimum cost intervention design, we can create a minimum cost intervention design. Further, the reduction is approximation preserving.

In practice, it can sometimes be difficult to intervene on a large number of variables. A variant of intervention design of interest is when every intervention can only involve $k$ variables. For this problem, we want our interventions to be a $k$-sparse graph separating system.

**Definition 4** ($k$-Sparse Graph Separating System). Given an undirected graph $G = (V, E)$, a $k$-*sparse graph separating system* of size $m$ is a collection of $m$ subsets of vertices $\mathcal{I} = \{I_1, I_2, \ldots, I_m\}$ such that all subsets $I_i$ satisfy $|I_i| \leq k$ and every edge is cut at least once, that is, $\bigcup_{I \in \mathcal{I}} \delta(I) = E$.

We consider two optimization problems related to $k$-sparse graph separating systems. In the first one we want to find a graph separating system of minimum size.

**Definition 5** (Minimum Size $k$-Sparse Intervention Design). Given a chordal graph $G = (V, E)$ and a sparsity constraint $k$, the *minimum size $k$-sparse intervention design problem* is to find a $k$-sparse graph separating system for $G$ of minimum size, that is, we want to minimize the cost

$$\text{cost}(\mathcal{I}) = |\mathcal{I}|.$$

For the next problem, we want to find the $k$-sparse intervention design of minimum cost where there is a cost to intervene on every variable.

**Definition 6** (Minimum Cost $k$-Sparse Intervention Design). Given a chordal graph $G = (V, E)$, a set of weights $w_v$ for all $v \in V$, a sparsity constraint $k$, and a size constraint $m$, the *minimum cost $k$-sparse intervention design problem* is to find a $k$-sparse graph separating system $\mathcal{I}$ of size $m$ that minimizes the cost

$$\text{cost}(\mathcal{I}) = \sum_{I \in \mathcal{I}} \sum_{v \in I} w_v.$$

## 3  Related Work

One problem of interest is to find the intervention design with the smallest number of interventions. Eberhardt et al. [2] established that $\lceil \log n \rceil$ is sufficient and nessesary in the worst case. Eberhardt [3] established that graph separating systems are necessary across all graphs (the example he used is the complete graph). Hauser and Bühlmann [10] establish the connection between graph colorings and intervention designs by using the key observation of Mao-Cheng [22] that graph colorings can be used to construct graph separating systems, and vise-versa. This leads to the requirement and sufficiency of $\lceil \log(\chi) \rceil$ experiments where $\chi$ is the chromatic number of the graph.

Since graph coloring can be done efficiently for chordal graphs, we can efficiently create a minimum size intervention design when given as input a chordal skeleton. Similarly, if we are given as input an arbitrary graph, perhaps due to side information on some edge directions, it is NP-hard to find a minimum size intervention design [13, 22].

Hu et al. [12] proposed a randomized algorithm that requires only $O(\log \log n)$ experiments and learns the causal graph with high probability.

Closer to our setup, Hyttinen et al. [13] considers a special case of minimum cost intervention design problem when every vertex has cost 1 and the input is the complete graph. They were able to optimally solve this special case. Kocaoglu et al. [19] was the first to formalize the minimum cost intervention design problem on general chordal graphs and the relationship to its coloring variant. They used the coloring variant to develop a greedy algorithm that finds a maximum weighted independent set and colors this set with the available color with the lowest weight. However their work did not establish approximation guarantees on this algorithm and it is not clear how many iterations the greedy algorithm needs to fully color the graph—we address these issues in this paper. Further it was unknown until our work that the minimum cost intervention design problem is NP-hard.

There has been a lot of prior work when every intervention is constrained to be of size at most $k$. Eberhardt et al. [2] was the first to consider the minimum size $k$-sparse intervention design problem and established sufficient conditions on the number of interventions needed for the complete graph. Hyttinen et al. [13] showed how $k$-sparse separating system constructions can be used for intervention designs on the complete graph using the construction of Katona [17]. They establish the necessary and sufficient number of $k$-sparse interventions needed to learn all causal directions in the complete graph. Shanmugam et al. [31] illustrate that for the complete graphs separating systems are necessary even under the constraint that each intervention has size at most $k$. They also identify an information theoretic lower bound on the necessary number of experiments and propose a new optimal $k$-sparse separating system construction for the complete graph. To the best of our knowledge there has been no graph dependent bounds on the size of a $k$-sparse graph separating systems until our work.

Ghassami et al. considered the dual problem of maximizing the number of learned causal edges for a given number of interventions [7]. They show that this problem is a submodular maximization problem when only interventions involving a single variable are allowed. We note that their connection to submodularity is different than the one we discover in our work.

Graph coloring has been extensively studied in the literature. There are various versions of graph coloring problem. We identify a connection of the minimum cost intervention design problem to the *general optimum cost chromatic partition problem* (GOCCP). GOCCP is a graph coloring problem where there are $t$ colors and a cost $\gamma_{vi}$ to color vertex $v$ with color $i$. It is a more general version of the minimum cost intervention design problem. Jansen [16] established that for graphs with bounded treewidth $r$, the GOCCP can be solved exactly in time $O(t^r n)$. This implies that for graphs with maximum degree $\Delta$ we can solve the minimum cost intervention design problem exactly in time $O(2^{m\Delta}n)$. Note that $m$ is at least $\log \Delta$ and can be as large as $\Delta$, thus this algorithm is not practical even for $\Delta = 12$.

## 4  Hardness of Minimum Cost Intervention Design

In this section, we show that the minimum cost intervention design problem is NP-hard.

We assume that the input graph is chordal, since it is obtained as an undirected component of a causal graph skeleton. We note that every chordal graph can be realized by this process.

**Proposition 7.** *For any undirected chordal graph $G$, there is a causal graph $D$ such that the essential graph $\mathcal{E}(D) = G$.*

Thus every chordal graph is the undirected subgraph of the essential graph for some causal DAG. This validates the problem definition of the minimum cost intervention design as any chordal graph can be given as input. We now state our hardness result.

**Theorem 8.** *The minimum cost intervention design problem is NP-hard.*

Please see Appendix D for the proof. Our proof is based on the reduction from numerical 3D matching to a graph coloring problem that is more general than the minimum cost intervention problem on interval graphs by Kroon et al. [21]. Our hardness proof holds even if the vertex costs are all equal to 1 and the input graph is an interval graph, which is a subset of chordal graphs that often have efficient algorithms for problems that are hard in general graphs.

It it worth comparing to complexity results on related minimum size intervention design problem. The minimum size intervention design problem on a graph can be solved by finding a minimum coloring on the same graph [22, 10]. For chordal graphs, graph coloring can be solved efficiently so the minimum size intervention design problem can also be solved efficiently. In contrast, the

minimum cost intervention design problem is NP-hard, even on chordal graphs. Both problems are hard on general graphs, which can be due to side information.

## 5 Approximation Guarantees for Minimum Cost Intervention Design

Since the input graph is chordal, we can find the maximum weighted independent sets in polynomial time using Frank's algorithm [4]. Further, a chordal graph remains chordal after removing a subset of the vertices. The authors of [19] use these facts to construct a greedy algorithm for this weighted coloring problem. Let $G_0 = G$. On iteration $t$, find the maximum weighted independent set in $G_t$ and assign these vertices the available color with the smallest cost. Then let $G_{t+1}$ be the graph after removing the colored vertices from $G_t$. Repeat this until all vertices are colored. Convert the coloring to a graph separating system and return this design.

One issue with this algorithm is it is not clear how many iterations the greedy algorithm will utilize until the graph is fully colored. This is important as we want to satisfy the size constraint on the graph separating system. To reduce the number of colors in the graph, we introduce a *quantization* step to reduce the number of iterations the greedy algorithm requires to completely color the graph. In Figure 3 of Appendix A, we see an example of a (non-chordal) graph where without quantization the greedy algorithm requires $n/2$ colors but with quantization it only requires $4$ colors.

Specifically, we first find the maximum independent set of the input graph and remove it. We then find the maximum cost vertex of the new graph with weight $w_{\max}$. For all vertices $v$ in the new graph, we replace the cost $w_v$ with $\lfloor \frac{w_v n^3}{w_{\max}} \rfloor$. See Algorithm 1 for pseudocode describing our algorithm.

The reason we first remove the maximum independent set before quantizing is because the maximum independent set will be colored with a color of weight $0$, and thus not contribute to the cost. We want the quantized costs to not be arbitrarily far from the original costs, except for the vertices that are not intervened on. For example, if there is a vertex with a weight of infinity, we will never intervene on it. However if we were to quantize it the optimal solution to the quantized problem can be arbitrarily far from the true optimal solution. Our method of quantization will allow us to show that a good solution to the quantized weights is also a good solution to the true weights.

---

**Algorithm 1** Greedy Coloring Algorithm with Quantization

---

    Input: A chordal graph $G = (V, E)$, positive integral weights $w_i$ for all $i \in V$.
    Quantize the vertex weights:

    $S_0 \leftarrow$ maximum weighted independent set of G
    $w_{\max} \leftarrow \max_{i \in V \setminus S_0} w_i$
    $w_i \leftarrow \lfloor \frac{w_i n^3}{w_{\max}} \rfloor$

    Greedy weighted coloring algorithm:

    Assign $S_0$ color $0$
    $G_1 \leftarrow G - S_0$
    $t \leftarrow 1$
    while $G_t$ is not empty:
        $S_t \leftarrow$ maximum weight independent set of $G_t$
        color all vertices of $S_t$ with the color $t$
        $G_{t+1} \leftarrow G_t - S_t$
        $t \leftarrow t + 1$
    convert the coloring of $G$ to a graph separating system $\mathcal{I}$
    return $\mathcal{I}$

---

We now state our main theorem, which guarantees that the greedy algorithm with quantization will return a solution that is a $(2 + \varepsilon)$-approximation from the optimal solution while only using $\log \chi + O(\log \log n)$ interventions. Our algorithm thus returns a good solution to the minimum cost intervention design problem whenever the allowed number of interventions $m \geq \log \chi + O(\log \log n)$. Note that $m \geq \log \chi$ is required for there to exist any graph separating system.

**Theorem 9.** *If the number of interventions $m$ satisfies $m \geq \log \chi + \log \log n + 5$, then the greedy coloring algorithm with quantization for the minimum cost intervention design problem creates a*

*graph separating system* $\mathcal{I}_{\text{greedy}}$ *such that*

$$\text{cost}(\mathcal{I}_{\text{greedy}}) \leq (2 + \varepsilon)\text{OPT},$$

*where* $\varepsilon = \exp(-\Omega(m)) + n^{-1}$.

See Appendix B for the proof of the theorem. We present a brief sketch of our proof.

To show that the greedy algorithm uses a small number of colors, we first define a submodular, monotone, and non-negative function such that every vertex has been colored if and only if this particular submodular function is maximized. This is an instance of the *submodular cover* problem. Wolsey established that the greedy algorithm for the submodular cover problem returns a set with cardinality that is close to the optimal cardinality solution when the values of the submodular function are bounded by a polynomial [35]. This is why we need to quantize the weights.

To show that the greedy algorithm returns a solution with small value, we first define a new class of functions which we call *supermodular chain functions*. We then show that the minimum cost intervention design problem is an instance of a supermodular chain function. Using result on submodular optimization from [23, 20] and some nice properties of the minimum cost intervention design problem, we are able to show that the greedy algorithm returns an approximately optimal solution.

To relate the quantized weights back to the original weights, we use an analysis that is similar to the analysis used to show the approximation guarantees of the knapsack problem [14].

Finally, we remark how our algorithm will perform when there are vertices with infinite cost. These vertices can be interpreted as variables that cannot be intervened on. If these variables form an independent set, then they can be colored with the color of weight zero. We can maintain our theoretical guarantees in this case, since our quantization procedure first removes the maximum weight independent set. If the variables with infinite cost do not form an independent set, then no valid graph separating system has finite cost.

## 6   Algorithms for $k$-Sparse Intervention Design Problems

We first establish a lower bound for how large a $k$-sparse graph separating system must be for a graph $G$ based on the size of the smallest vertex cover of the graph $\tau$.

**Proposition 10.** *For any graph $G$, the size of the smallest $k$-sparse graph separating system $m_k^*$ satisfies $m_k^* \geq \frac{\tau}{k}$, where $\tau$ is the size of the smallest vertex cover in the graph $G$.*

See Appendix C for the proof.

---

**Algorithm 2** Algorithm for Min Size and Unweighted Min Cost $k$-Sparse Intervention Design

---

Input: A chordal graph $G$, a sparsity constraint $k$.
$S \leftarrow$ minimum size vertex cover of $G$.
$G_S \leftarrow$ induced graph of $S$ in $G$.
Find an optimal coloring of $G_S$.
Split the color classes of $G_S$ into size $k$ intervention sets $I_1, I_2, \ldots, I_m$.
Return $\mathcal{I} = \{I_1, I_2, \ldots, I_m\}$.

---

We use Algorithm 2 to find a small $k$-sparse graph separating system. It first finds the minimum cardinality vertex cover $S$. It then finds an optimal coloring of the graph induced with the vertices of $S$. It then partitions the color class into independent sets of size $k$ and performs an intervention for each of these partitions. Since the set of vertices not in a vertex cover is an independent set, this is a valid $k$-sparse graph separating system.

When the sparsity $k$ and the maximum degree $\Delta$ are small, Algorithm 2 is nearly optimal. Using Proposition 10, we can establish the following approximation guarantee on the size of the graph separating system created.

**Theorem 11.** *Given a chordal graph $G$ with maximum degree $\Delta$, Algorithm 2 finds a $k$-sparse graph separating system of size $m_k$ such that*

$$m_k \leq \left(1 + \frac{k(\Delta + 1)\Delta}{n}\right)\text{OPT},$$

*where* OPT *is the size of the smallest $k$-sparse graph separating system.*

See Appendix C for the proof. If the sparsity constraint $k$ and the maximum degree of the graph $\Delta$ both satisfy $k, \Delta = o(n^{1/3})$, then Theorem 11 implies that we have a $1 + o(1)$ approximation to the optimal solution.

One interesting aspect of Algorithm 2 is that every vertex is only intervened on once and the set of elements not intervened on is the maximum cardinality independent set. By a similar argument to Theorem 2 of [19], we have that this algorithm is optimal in the unweighted case.

**Corollary 12.** *Given an instance of the minimum cost $k$-sparse intervention design problem with chordal graph $G$ with maximum degree $\Delta$ and vertex cover of size $\tau$, sparsity constraint $k$, size constraint $m \geq \frac{\tau}{k}(1 + \frac{k(\Delta+1)\Delta}{n})$, and all vertex weights $w_v = 1$, Algorithm 2 returns a solution with optimal cost.*

We show one way to extend Algorithm 2 to the weighted case. There is a trade off between the size and the weight of the independent set of vertices that are never intervened on. We can trade these off by adding a penalty $\lambda$ to every vertex weight, i.e., the new weight $w_v^\lambda$ of a vertex $v$ is $w_v^\lambda = w_v + \lambda$. Larger values of $\lambda$ will encourage independent sets of larger size. See Algorithm 3 for the pseudocode describing this algorithm. We can run Algorithm 3 for various values of $\lambda$ to explore the trade off between cost and size.

---

**Algorithm 3** Algorithm for Weighted Min Cost $k$-Sparse Intervention Design

Input: chordal graph $G$, sparsity constraint $k$, vertex weights $w_v$, penalty parameter $\lambda$
$S \leftarrow$ minimum weight vertex cover $S$ using weights $w_v^\lambda = w_v + \lambda$.
$G_S \leftarrow$ induced graph of $S$ in $G$.
Find an optimal coloring of $G_S$.
Split the color classes of $G_S$ into size $k$ intervention sets $I_1, I_2, \ldots, I_m$.
Return $\mathcal{I} = \{I_1, I_2, \ldots, I_m\}$.

---

## 7 Experiments

We generate chordal graphs following the procedure of [31], however we modify the sampling algorithm so that we can control the maximum degree. First we order the vertices $\{v_1, v_2, \ldots, v_n\}$. For vertex $v_i$ we choose a vertex from $\{v_{i-b}, v_{i-b+1}, \ldots, v_{i-1}\}$ uniformly at random and add it to the neighborhood of $v_i$. We then go through the vertices $\{v_{i-b}, v_{i-b+1}, \ldots, v_{i-1}\}$ and add them to the neighborhood of $v_i$ with probability $\frac{d}{b}$. We then add edges so that the neighbors of $v_i$ in $\{v_1, v_2, \ldots, v_{i-1}\}$ form a clique. This is guaranteed to be a connected chordal graph with maximum degree bounded by $2b$.

In our first experiment we compare the greedy algorithm to two other algorithms. One first assigns the maximum weight independent set the weight 0 color, then finds a minimum coloring of the remaining vertices, sorts the independent sets by weight, then assigns the cheapest colors to the independent sets of the highest weight. The other algorithm finds the optimal solution with integer programming using the Gurobi solver[8]. The integer programming formulation is standard (see, e.g., [1]).

We compare the cost of the different algorithms when we (a) adjust the number of vertices while maintaining the average degree and (b) adjust the average degree while maintaining the number of vertices. We see that the greedy coloring algorithm performs almost optimally. We also see that it is able to find a proper coloring even with only $m = 5$ interventions and no quantization. See Figure 1 for the complete results.

In our second experiment we see how Algorithm 3 allows us to trade off the number of interventions and the cost of the interventions in the $k$-sparse minimum cost intervention design problem. See Figure 2 for the results.

Finally, we observe the empirical running time of the greedy algorithm. We generate graphs on $10,000$ vertices with maximum degree 20 and have 5 interventions. The greedy algorithm terminates in 5 seconds. In contrast, the integer programming solution takes 128 seconds using the Gurobi solver [8].

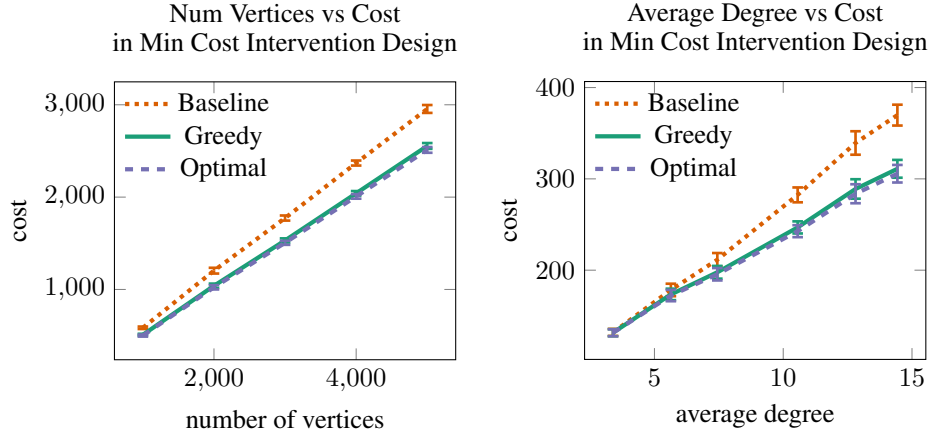

(a) We adjust the number of vertices. The average degree stays close to 10 for all values of the number of vertices.

(b) The number of vertices are fixed at 500. We adjust the sparsity parameter in the graph generator to see how the algorithms perform for varying graph densities.

Figure 1: We generate random chordal graphs such that the maximum degree is bounded by 20. The node weights are generated by the heavy-tailed Pareto distribution with scale parameter 2.0. The number of interventions $m$ is fixed to 5. We compare the greedy algorithm to the optimal solution and the baseline algorithm mentioned in the experimental setup. We see that the greedy algorithm is close to optimal and outperforms the baseline. We also see that the greedy algorithm is able to find a solution with the available number of colors, even without quantization.

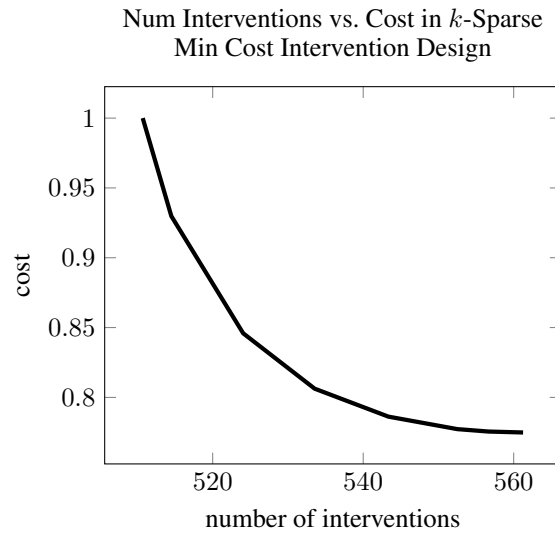

Figure 2: We sample graphs of size 10000 such that the maximum degree is bounded by 20 and the average degree is 3. We draw the weights from the heavy-tailed Pareto distribution with scale parameter 2.0. We restrict all interventions to be of size 10. We adjust the penalty parameter in Algorithm 3 to see how the size of the $k$-sparse graph separating system relates to the cost. Costs are normalized so that the largest cost is 1.0. We see that with 561 interventions we can achieve a cost of 0.78 compared to a cost of 1.0 with 510 interventions. Our lower bound implies that we need 506 interventions on average.

## Acknowledgments

This material is based upon work supported by the National Science Foundation Graduate Research Fellowship under Grant No. DGE-1110007. This research has been supported by NSF Grants CCF 1422549, 1618689, DMS 1723052, CCF 1763702, ARO YIP W911NF-14-1-0258 and research gifts by Google, Western Digital, and NVIDIA.

## Footnotes

[1] To be more precise, parent nodes are said to directly cause a variable whereas ancestors cause indirectly through parents. In this paper, we will not make this distinction since we do not use indirect causal relations for graph discovery.

[2] It is known that the edges identified in a chordal component of the skeleton do not help identify edges in another component [9].Thus, each chordal component learning task can be treated as an individual problem.

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
