[Supplementary Material]

# Appendix

## A  Example Graph Where Quantization Helps Greedy

Figure 3: When there are very large weights, the greedy algorithm may require a lot of colors to terminate, even on graphs with a small chromatic number. For this graph, the largest independent set is the top two vertices, followed by the next two, and so on. The greedy algorithm will color all these pairs of vertices a different color, which is $\frac{n}{2}$ colors. However after quantization the the greedy algorithm will only use 4 colors.

## B  Proof of Approximation Guarantees of the Quantized Greedy Algorithm

In this section we prove our approximation guarantee of using the quantized greedy algorithm for minimum cost intervention design.

**Theorem 9.** *If the number of interventions $m$ satisfies $m \geq \log \chi + \log \log n + 5$, then the greedy algorithm with quantization for the minimum cost intervention design problem creates a graph separating system $\mathcal{I}_{\mathrm{greedy}}$ such that*

$$\mathrm{cost}(\mathcal{I}_{\mathrm{greedy}}) \leq (2 + \varepsilon)\mathrm{OPT},$$

*where $\varepsilon = \exp(-\Omega(m)) + n^{-1}$.*

### B.1  Submodularity Background

Our proof uses several results from submodularity. A set function $F$ over a ground set $V$ is a function that takes as input a subset of $V$ and outputs a real number. We say that the function $F$ is *submodular* if for all $v \in V$ and $A \subseteq B \subseteq V \setminus \{v\}$ the function satisfies the diminishing returns property

$$F(A \cup \{v\}) - F(A) \geq F(B \cup \{v\}) - F(B).$$

We say that the function $F$ is monotone if for all $A \subseteq B \subseteq V$ we have that $F(A) \leq F(B)$. We say that $F$ is *non-negative* if for all $A \subseteq V$ we have that $F(A) \geq 0$.

One classic problem in submodular optimization is finding a set $A$ with cardinality constraint $|A| \leq k$ that maximizes a submodular, monotone, and non-negative function $F$. The greedy algorithm starts with the emptyset $A_0 = \emptyset$, selects the item $v_{k+1} = \arg\max_{v \in V} F(A_k \cup \{v\}) - F(A_k)$. It then updates $A_{k+1} = A_k \cup \{v_{k+1}\}$.

The classic result of Nemhauser and Wolsey established that the greedy algorithm is a $(1 - 1/e)$-approximation algorithm to the optimal [23]. Krause and Golovin generalized this to show that if the greedy algorithm selects $\lceil Ck \rceil$ elements for some positive value $C$, then it is a $(1 - e^{-C})$-approximation to the optimal solution of size $k$.

**Theorem 13** ([23, 20]). *Given a submodular, monotone, and non-negative function $F$ over a ground set $V$ and a cardinality constraint $k$, let $\mathrm{OPT}$ be defined as*

$$\mathrm{OPT} = \max_{A \subseteq V : |A| \leq k} F(A).$$

*If the greedy algorithm for this problem runs for $\lceil Ck \rceil$ iterations, for some positive value $C$, it returns a set $A_{\text{greedy}}^{Ck}$ such that*

$$F(A_{\text{greedy}}^{Ck}) \geq (1 - e^{-C})\text{OPT}.$$

Another important problem in submodular function optimization is the *submodular set cover* problem, which is a generalization of the set cover problem. Given a submodular, monotone, and non-negative function $F$ that maps a subsets of a ground set $V$ to integers, we want to find a set $A$ of minimum cardinality such that $F(A) = F(V)$. The greedy algorithm is a natural approach to solve this problem: we run greedy iterations until the set satisfies the submodular set cover constraint. Let $w_{\max} = \arg\max_{v \in V} F(\{v\})$. Wolsey established that the cardinality of the set returned by the greedy algorithm is a $1 + \ln w_{\max}$ approximation to the minimum cardinality solution [35].

**Theorem 14** ([35])**.** *Given a submodular, monotone, and non-negative function $F$ that maps subsets of a ground set $V$ to integers, let* OPT *be defined as*

$$\text{OPT} = \min_{A \subseteq V : F(A) = F(V)} |A|.$$

*Let $w_{\max} = \arg\max_{v \in V} F(\{v\})$. The greedy algorithm for this problem returns a set $A_{\text{greedy}}$ such that*

$$|A_{\text{greedy}}| \leq (1 + \ln w_{\max})\text{OPT}.$$

## B.2  Bound on the Quantized Greedy Algorithm solution size

In this section we show that after $\chi(2 + 5\ln n) + 1$ rounds the greedy algorithm with quantization will have colored every vertex in the graph. Since the number of possible colors in a graph separating system of size $m$ is $2^m$, this implies that when $m \geq \log\chi + \log\log n + 4$, there are enough colors for the greedy algorithm to fully color the graph.

**Lemma 15.** *If the intervention size is $m \geq \log\chi + \log\log n + 4$, then the greedy algorithm will terminate using at most $2^m$ colors.*

*Proof.* The greedy algorithm first colors the maximum weight independent, using 1 color. We will denote the remaining graph by $G = (V, E)$.

The weights of the remaining vertices are quantized to integers such that the maximum weight is bounded by $n^3$. Let $\mathcal{A}$ be the set of all independent sets in $G$. The maximum weight of an independent set in $G$ is bounded by $n^4$. Let $W$ be the function that takes a set of independent sets $A \subseteq \mathcal{A}$ and outputs the value

$$W(A) = \sum_{v \in \bigcup_{a \in A} a} w_v,$$

that is, it takes a set of independent sets and return the sum of the vertices in their union. It can be verified that this function is submodular, monotone, and non-negative.

We will assume for now that the weights are all positive. If we have a set of independent sets $A$ such that $W(A) = W(\mathcal{A})$, then every vertex in the graph will have been covered. Since the minimum cardinality is $\chi$ and the maximum weight of an independent set is $n^4$, by Theorem 14, the greedy algorithm will terminate after $\chi(1 + 4\ln n)$ iterations.

To handle vertices of weight $0$, note that it is a set cover problem to cover the remaining vertices. Thus the greedy algorithm will need no more that $\chi(1 + \ln n)$ colors to color the remaining vertices, using a total number of colors $\chi(2 + 5\ln n) + 1$. □

We have the following corollary by noting that adding an extra intervention doubles the number of allowed colors.

**Corollary 16.** *If the intervention size is $m \geq \log\chi + \log\log n + 5$, then the greedy algorithm will terminate using at most $2^m$ colors such that all color vectors $c$ have weight $\|c\|_1 \leq \lceil \frac{m}{2} \rceil$.*

## B.3 Submodular and Supermodular Chain Problem

In this section we define two new types of submodular optimization problem, which we call the submodular chain problem and the supermodular chain problem. We will use these in our proof of the approximation guarantees of the greedy algorithm with quantization.

**Definition 17.** Given integers $k_1, k_2, \ldots, k_m$ and a submodular, monotone, and non-negative function $F$, over a ground set $V$, the *submodular chain problem* is to find sets $A_1, A_2, \ldots, A_m \subseteq V$ such that $|A_i| \leq k_i$ that maximizes

$$\sum_{i=1}^{m} F(A_1 \cup A_2, \cup \cdots \cup A_i).$$

Throughout this section we will assume that $m$ is an even number.

The greedy algorithm for this problem will first choose the set $A_1$ of cardinality $k_1$ that maximizes $F(A_1)$. It will then choose the set $A_2$ of cardinality $k_2$ that maximizes $F(A_1 \cup A_2)$. It will continue this process until all $A_i$ are chosen.

Note by using the greedy algorithm and Theorem 13 we can obtain a $(1 - 1/e)$ approximation to this problem. However we will instead use the following guarantee.

**Lemma 18.** *Let $A_1^*, A_2^*, \ldots, A_m^*$ be the optimal solution to the submodular chain problem. Suppose that for all $1 \leq p \leq m/2 - 1$ we have that $\sum_{i=1}^{2p} k_i \geq C \sum_{i=1}^{p} k_i$. Also assume that $F(A_1 \cup A_2 \cup \cdots \cup A_m) = F(V)$. Then the greedy algorithm for the submodular chain problem returns set $A_1, A_2, \ldots, A_m$ such that*

$$\sum_{i=0}^{m} F(A_1 \cup A_2 \cup \cdots A_{2i}) \geq F(V) + 2(1 - e^{-C}) \sum_{i=0}^{m/2-1} F(A_1^* \cup A_2^* \cup \cdots A_i^*)$$

*Proof.* Since $\sum_{p=1}^{2i} k_p \geq C \sum_{p=1}^{i} k_p$, by Theorem 13 we have that

$$F(A_1 \cup A_2 \cup \cdots A_{2p}) \geq (1 - e^{-C})F(A_1^* \cup A_2^* \cup \cdots \cup A_p^*).$$

We thus have

$$\sum_{i=0}^{m/2-1} F(A_1 \cup A_2 \cup \cdots \cup A_{2i}) \geq (1 - e^{-C}) \sum_{i=0}^{m/2-1} F(A_1^* \cup A_2^* \cup \cdots \cup A_i^*).$$

To conclude the proof, use the monotonicity of the submodular function $F$ to observe that

$$\sum_{i=0}^{m} F(A_1 \cup A_2 \cup \cdots \cup A_i) = F(A_1 \cup A_2 \cup A_m)$$

$$+ \sum_{i=0}^{m/2-1} F(A_1 \cup A_2 \cup \cdots \cup A_{2i}) + F(A_1 \cup A_2 \cup \cdots \cup A_{2i+1})$$

$$= F(V) + \sum_{i=0}^{m/2-1} F(A_1 \cup A_2 \cup \cdots \cup A_{2i}) + F(A_1 \cup A_2 \cup \cdots \cup A_{2i+1})$$

$$\geq F(V) + 2 \sum_{i=0}^{m/2-1} F(A_1 \cup A_2 \cup \cdots \cup A_{2i}).$$

$\square$

We define the supermodular chain problem similarly.

**Definition 19.** Given integers $k_1, k_2, \ldots, k_m$ and a submodular, monotone, and non-negative function $F$, over a ground set $V$, the *supermodular chain problem* is to find sets $A_1, A_2, \ldots, A_m \subseteq V$ such that $|A_i| \leq k_i$ that minimizes

$$\sum_{i=0}^{m} F(V) - F(A_1 \cup A_2, \cup \cdots \cup A_i).$$

We establish the following guarantee for the greedy algorithm on the supermodular chain problem.

**Lemma 20.** *Let $A_1^*, A_2^*, \ldots, A_m^*$ be the optimal solution to the supermodular chain problem. Suppose that for all $1 \leq p \leq m/2 - 1$ we have that $\sum_{i=1}^{2t} k_i \geq C \sum_{i=1}^{t} k_i$. Also assume that $F(A_1 \cup A_2 \cup \cdots \cup A_m) = F(V)$. Then the greedy algorithm for the supermodular chain problem returns set $A_1, A_2, \ldots, A_m$ such that*

$$\sum_{i=0}^{m} F(V) - F(A_1 \cup A_2 \cup \cdots A_i) \leq e^{-C} m F(V) + 2 \sum_{i=0}^{m} F(A_1^* \cup A_2^* \cup \cdots A_i^*)$$

*Proof.* Starting from Lemma 18, we have that

$$(m+1)F(V) - \sum_{i=0}^{m} F(A_1 \cup A_2 \cup \cdots A_{2i}) \leq mF(V) - 2(1-e^{-C}) \sum_{i=0}^{m/2-1} F(A_1^* \cup A_2^* \cup \cdots A_i^*)$$

$$\leq e^{-C} m F(V) + mF(V) - 2 \sum_{i=0}^{m/2-1} F(A_1^* \cup A_2^* \cup \cdots A_i^*)$$

$$= e^{-C} m F(V) + 2 \sum_{i=0}^{m/2-1} F(V) - F(A_1^* \cup A_2^* \cup \cdots A_i^*).$$

Using the monotonicity of the submodular function $F$, we can continue with

$$e^{-C} m F(V) + 2 \sum_{i=0}^{m/2-1} F(V) - F(A_1^* \cup A_2^* \cup \cdots A_i^*) \leq e^{-C} m F(V) + 2 \sum_{i=0}^{m} F(V) - F(A_1^* \cup A_2^* \cup \cdots A_i^*)$$

and conclude that

$$\sum_{i=0}^{m} F(V) - F(A_1 \cup A_2 \cup \cdots A_{2i}) = (m+1)F(V) - \sum_{i=0}^{m} F(A_1 \cup A_2 \cup \cdots A_{2i})$$

$$\leq e^{-C} m F(V) + 2 \sum_{i=0}^{m} F(V) - F(A_1^* \cup A_2^* \cup \cdots A_i^*)$$

$\square$

### B.4  Proof of quantized greedy algorithm approximation guarantees

For simplicity, we will assume that the number of interventions $m$ is divisible by $4$.

We will need the following lemma, which can be proved by standard binomial approximations.

**Lemma 21.** *If $m$ and $t$ are integers such that $t \leq \frac{m}{4}$, we have*

$$\sum_{i=1}^{2t} \binom{m}{i} \geq \Omega(m) \left(1 + \sum_{i=1}^{t} \binom{m}{i}\right).$$

We use the following technical lemma to prove our approximation guarantee. We defer the proof to Section B.5.

**Lemma 22.** *Let $A^*$ be the optimal solution to the coloring problem. Let $A^+$ be the optimal solution to the coloring problem when we force it to color the maximum weighted independent set with the weight $0$ color, but allow it an extra color of weight $1$. That is, it can color $m + 1$ independent sets with a color of weight $1$, rather than the usual $m$ independent sets. We have*

$$\text{cost}(A^+) \leq \text{cost}(A^*).$$

We can now show that the quantized greedy algorithm is a good approximation to the optimal solution to the quantized problem.

**Lemma 23.** *Suppose all the weights in the graph that are not in the maximum weight independent set are bounded by $n^3$. Then if the number of interventions $m$ satisfies $m \geq \log \chi + \log \log n + 5$ the greedy coloring algorithm returns a solution $\mathcal{I}$ of cost*

$$\text{cost}(\mathcal{I}) \leq (2 + \exp(-\Omega(m)))\text{OPT}.$$

*Proof.* Let $\mathcal{A}$ be the set of all independent sets in $G$. Let $W$ be the function that takes a set of independent sets $A \subseteq \mathcal{A}$ and outputs the value

$$W(A) = \sum_{v \in \bigcup_{a \in A} a} w_v,$$

that is, it takes a set of independent sets and return the sum of the vertices in their union. It can be verified that this function is submodular, monotone, and non-negative.

A feasible solution to the coloring variant of the minimum cost intervention design problem is a coloring that maps vertices to color vectors $\{0, 1\}^m$. The colors $c$ with weight $i$ are the coloring vectors $c$ such that $\|c\|_1 = i$. We can describe a feasible solution to the coloring variant of the minimum cost intervention design by $A_0, A_1, A_2, \ldots, A_m$, where $A_i$ is the set of independent sets that are colored with a coloring vector of weight $i$.

One simplifying assumption is that the optimal solution $A^*$ and the greedy solution $A$ both use the color of weight $0$ to color the maximum weight independent set. By Lemma 22 this assumption is valid if we allow the optimal solution to use an additional color of weight $1$. We just need to show the approximation guarantee on the sets $A_1, A_2, \ldots, A_m$.

We can calculate the cost of a feasible solution of the minimum cost intervention design problem by

$$\text{cost}(A_1, \ldots, A_m) = \sum_{i=0}^{m} W(\mathcal{A}) - W(A_1 \cup A_2 \cup \cdots A_m),$$

where $|A_i| \leq \binom{m}{i}$. This is an instance of the supermodular chain problem.

Using Corollary 16, the greedy algorithm will terminate only using colors of weight at most $m/2$, so we only need to show optimality of the sets $A_1, A_2, \ldots, A_{m/2}$. By Lemma 21, we have that the number of colors of weight at most $2t$ is a factor of $\Omega(m)$ more than the number of colors the optimal solution uses of weight at most $t$, even after including the extra color given to the optimal solution. By Lemma 20 and using the monotonicity of $W$, we have that

$$
\begin{aligned}
\text{cost}(\mathcal{I}_{\text{greedy}}) &= \text{cost}(A_1, A_2, \ldots, A_{m/2}) \\
&= \sum_{i=0}^{m/2} W(\mathcal{A}) - W(A_1 \cup A_2 \cup \cdots A_i) \\
&\leq e^{-\Omega(m)} \frac{m}{2} F(\mathcal{A}) + 2 \sum_{i=0}^{m/2} W(\mathcal{A}) - W(A_1^* \cup A_2^* \cup \cdots A_i^*) \\
&\leq e^{-\Omega(m)} \frac{m}{2} F(\mathcal{A}) + 2 \sum_{i=0}^{m} W(\mathcal{A}) - W(A_1^* \cup A_2^* \cup \cdots A_i^*) \\
&= e^{-\Omega(m)} \frac{m}{2} W(\mathcal{A}) + 2\text{OPT}.
\end{aligned}
$$

To conclude, observe that $\text{OPT} \geq W(\mathcal{A})$, since every vertex not in the maximum weighted independent set is colored with a color of weight at least $1$. $\square$

Lemma 23 shows an approximation guarantee of the quantized greedy algorithm to the quantized optimal solution. To relate the quantized greedy algorithm to the true optimal solution, we use the following lemma, which we prove in Section B.5.

**Lemma 24.** *Suppose an intervention design $\mathcal{I}$ is an $\alpha$-approximation solution to the optimal solution to the quantized problem. Then it is an $(\alpha + n^{-1})$-approximation to the optimal solution to the original problem.*

With Lemma 24, we can conclude the proof of Theorem 9.

## B.5 Proof of Technical Lemmas

**Lemma 22.** *Let $A^*$ be the optimal solution to the coloring problem. Let $A^+$ be the optimal solution to the coloring problem when we force it to color the maximum weighted independent set with the weight $0$ color, but allow it an extra color of weight $1$. That is, it can color $m + 1$ independent sets with a color of weight $1$, rather than the usual $m$ independent sets. We have*

$$\text{cost}(A^+) \leq \text{cost}(A^*).$$

*Proof.* Let $a_0^*$ and $a_0^+$ be the sets of vertices covered with the color of weight $0$ for $A^*$ and $A^+$, respectively. From the optimality of $a_0^+$ as a maximum weight independent set, we have

$$\sum_{i \in a_0^+ \setminus a_0^*} w_i \geq \sum_{i \in a_0^* \setminus a_0^+} w_i.$$

Consider a new coloring $A'$, also with an extra weight $1$ coloring, that uses $a_0^+$ as the set of vertices colored with the weight $0$ color, $a_0^* \setminus a_0^+$ as the set of vertices colored with the extra weight $1$ color, then does the same coloring as $A^*$, removing the vertices that are already colored.

The only vertices colored by $A'$ with a positive cost and different color than $A^*$ are $a_0^* \setminus a_0^+$, which are all colored with a cost of weight $1$. The only vertices colored by $A^*$ with a positive cost and different color than $A'$ are $a_0^+ \setminus a_0^*$. Let $c_v^*$ be the cost to color vertex $v$ using $A^*$. We can thus conclude

$$\text{cost}(A') - \text{cost}(A^*) = \sum_{v \in a_0^* \setminus a_0^+} w_v - \sum_{v \in a_0^+ \setminus a_0^*} c_v^* w_v$$

$$\leq \sum_{v \in a_0^* \setminus a_0^+} w_v - \sum_{v \in a_0^+ \setminus a_0^*} w_v \leq 0.$$

$\square$

**Lemma 24.** *Suppose an intervention design $\mathcal{I}$ is an $\alpha$-approximation solution to the optimal solution to the quantized problem. Then it is an $(\alpha + n^{-1})$-approximation to the optimal solution to the original problem.*

*Proof.* This is a modification of the proof of the FPTAS of the knapsack algorithm [14] (see also [34]).

Let $c^*$ be the optimal coloring in the original weights, $c'$ be the optimal coloring in the quantized weights, and $c$ be the approximate coloring.

Let $w_v$ be the true weight, and $w_v'$ be the quantized weight. Let $\mu = \frac{w_{\max}}{n^3}$. Since $w_v' = \lfloor \frac{w_v}{\mu} \rfloor$, we have and $w_v' \leq \frac{w_v}{\mu}$. We also have $\text{cost}(\mathcal{I}^*) \geq w_{\max}$.

We thus have

$$\text{cost}(\mathcal{I}) = \sum_{v \in V} \|c(v)\|_1 w_v$$

$$\leq \mu \sum_{v \in V} \|c(v)\|_1 (w_v' + 1)$$

$$= \mu \sum_{v \in V} \|c(v)\|_1 w_v' + \mu \sum_{v \in V} \|c(v)\|_1$$

$$\leq \alpha \mu \sum_{v \in V} \|c'(v)\| w_v' + \mu \sum_{v \in V} \|c(v)\|_1$$

Using the optimality of $c'$ in the quantized weights, we have

$$\alpha\mu \sum_{v \in V} \|c'(v)\|w'_v + \mu \sum_{v \in V} \|c(v)\|_1 \leq \alpha\mu \sum_{v \in V} \|c^*(v)\|_1 w'_v + \mu \sum_{v \in V} \|c(v)\|_1$$

$$\leq \alpha \sum_{v \in V} \|c^*(v)\|_1 w_v + \mu \sum_{v \in V} \|c(v)\|_1$$

$$= \alpha\text{OPT} + \mu \sum_{v \in V} \|c(v)\|_1$$

$$\leq \alpha\text{OPT} + \mu mn$$

$$= \alpha\text{OPT} + \frac{w_{\max}mn}{n^3}$$

$$\leq \alpha\text{OPT} + \frac{w_{\max}}{n}$$

$$\leq (\alpha + n^{-1})\text{OPT}.$$

$\square$

## C  Proof of Results on $k$-Sparse Intervention Design Problems

**Proposition 10.** *For any graph $G$, the size of the smallest $k$-sparse graph separating system $m_k^*$ satisfies $m_k^* \geq \frac{\tau}{k}$, where $\tau$ is the size of the smallest vertex cover in the graph $G$.*

*Proof.* Suppose that there exists a graph separating system $\mathcal{I}$ of size $m_k^* < \frac{\tau}{k}$. Note that the vertices in $S = \bigcup_{I \in \mathcal{I}} I$ form a vertex cover. The number of vertices in $S$ is $|S| \leq km_k^* < \tau$, contradicting the result that the smallest vertex cover has at least $\tau$ vertices. $\square$

**Theorem 11.** *Given a chordal graph $G$ with maximum degree $\Delta$, Algorithm 2 finds a $k$-sparse graph separating system of size $m_k$ such that*

$$m_k \leq \left(1 + \frac{k(\Delta + 1)\Delta}{n}\right) m_k^*,$$

*where $m_k^*$ is the size of the smallest $k$-sparse graph separating system.*

*Proof.* Given the vertices $S$ in the smallest vertex cover of the graph, we can color these vertices with $\Delta + 1$ colors. We can then partition each color class into $\frac{\tau}{k} + \Delta + 1$ independent sets of size $k$, as we have at most $\frac{\tau}{k}$ of size $k$ and at most $\Delta + 1$ sets that cannot be grouped into exactly $k$ vertices due to rounding errors.

Note that the size of the smallest vertex cover $\tau$ satisfies $\tau \geq \frac{n}{\Delta}$. We have

$$\Delta + 1 = \frac{k(\Delta+1)\Delta}{n} \frac{n}{k\Delta} \leq \frac{k(\Delta+1)\Delta}{n} \frac{\tau}{k} \leq \frac{k(\Delta+1)\Delta}{n} m_k^*.$$

Thus we use at most $\frac{\tau}{k} + \Delta + 1 \leq \left(1 + \frac{k(\Delta+1)\Delta}{n}\right) m_k^*$ interventions. $\square$

## D  Proof of NP-Hardness

We establish the following theorem in this section.

**Theorem 25.** *The minimum cost intervention design problem is NP-hard, even if every vertex has weight $1$ and the input graph is an interval graph.*

Theorem 8 follows immediately from Theorem 25.

First, we need to introduce the numerical three dimensional matching problem:

**Definition 26** (Numerical Three Dimensional Matching)**.** Given a positive integer $t$ and $3t$ rational numbers $a_i, b_i, c_i$ satisfying $\sum_{i=1}^{t} a_i + b_i + c_i = t$ and $0 < a_i, b_i, c_i < 1, \forall i \in [t]$, does there exist permutations $\rho, \sigma$ of $[t]$ such that $a_i + b_{rho(i)} + c_{\sigma(i)} = 1, \forall i \in [t]$?

The numerical three dimensional matching problem is known to be strongly NP-complete [5].

Kroon et al. [21] reduces the optimal cost chromatic partition problem on interval graphs to numerical three dimensional matching. The input to an instance of the optimal cost chromatic partitioning problem is a graph and a set of weighted colors. The cost to color a vertex with a given color is the weight of that color. The cost of a coloring is the sum of the coloring cost of each vertex. A solution to the problem is a valid coloring of minimum cost.

Kroon et al. show that the optimal cost chromatic partition problem is NP-hard, even if the input graph is an interval graph and color weights take four values: $0$, $1$, $2$, and $\Theta(n)$. They use the following construction for their reduction.

Suppose we are given an instance of the numerical three dimensional matching problem containing the number $a_i, b_i, c_i$ for $i \in [t]$. For $i, j \in [t]$, define rational numbers $A_i$, $B_j$, and $X_{ij}$ such that $4 < A_i < 5 < B_j < 6$ and $7 < X_{ij} < 9$. The following are the intervals of the graph used in [21] (see the original paper for an image):

| Interval | Occurrences | Clique ID |
|---|---|---|
| $(0, 1)$ | $t$ times | I |
| $(0, 3)$ | $t^2 - t$ times | I |
| $(0, A_i)$ | $\forall i \in [t], t - 1$ times | I |
| $(0, B_j)$ | $\forall j \in [t]$ | I |
| $(1, 2)$ | $t$ times | II |
| $(2, A_i)$ | for $\forall i \in [t]$ | III |
| $(3, B_j)$ | $\forall j \in [t], t - 1$ times | III |
| $(A_i, X_{i,j})$ | $\forall i, j \in [t]$ | IV |
| $(B_j, X_{i,j})$ | $\forall i, j \in [t]$ | IV |
| $(X_{i,j}, 10 + a_i + b_j)$ | $\forall i, j \in [t]$ | V |
| $(X_{i,j}, 14)$ | $\forall i, j \in [t]$ | V |
| $(11 - c_k, 13)$ | $\forall k \in [t]$ | VI |
| $(12, 14)$ | $t^2 - t$ | VII |
| $(13, 14)$ | $t$ times | VII |

They estabish that it is NP-complete to decide if there is a coloring of cost at most $11t^2 - 5t$ when there are $t$ colors of weight 0, $t^2 - t$ colors of weight 1, $t^2$ colors of weight 2, and all other colors of weight 3. However they omit the proof, so we include a proof here. We us the clique IDs we added in the definition of the interval graph.

*Proof.* If there is a solution to the numerical three dimensional matching problem, then there exists a coloring of cost at most $11t^2 - 5t$; see the original paper for the proof of this [21]. They also prove that if there is a coloring of cost at most $11t^2 - 5t$ that only uses the colors of weight 0, 1, and 2, then it can be used to construct a solution to the numerical three dimensional matching problem.

Now we show that if the coloring uses a color of weight 3, then it must have a cost strictly greater than $11t^2 - 5t$. Note that all the vertices with the same clique ID indeed do form a clique. Consider the subgraph containing all the vertices of the original graph, but only the edges between vertices with the same clique ID.

The optimal way to color a subgraph of size $k$ is to use one instance of the $k$ cheapest colors. From this, we can see that the optimal way to color the subgraph has a cost of $11t^2 - 5t$.

We can also see that any coloring of this subgraph that uses a color of weight 3 has a cost strictly larger than $11t^2 - 5t$. Since there is an available color of weight less than 3, if we swap the color of weight 3 with an available, cheaper color, the cost must decrease by at least 1. Since the coloring

after the switch cannot be lower than $11t^2 - 5t$, it must have been that the coloring before the switch was strictly larger than $11t^2 - 5t$.

Since a valid coloring for the original graph is a valid coloring for the subgraph, and the cost of a coloring of the original graph is the same as a cost of the coloring for the subgraph, we see that the cost of a coloring of the original graph that uses a color of weight 3 must have a cost strictly larger than $11t^2 - 5t$. $\qquad\square$

We also see that the problem still remains hard when there are $t$ colors of weight 1, $t^2 - t$ colors of weight 2, $t^2$ colors of weight 3, and all other colors of weight 4. This is because the cost of a coloring using these new colors is just an additive factor $n$ more than the original colors. Thus a coloring that minimizes the cost using these new colors also minimizes the cost using the original colors, and it is NP-complete to decide if there exists a coloring of cost $19t^2 - 3t$.

We will define another interval graph by adding the following intervals. Set $\varepsilon$, $\delta$ to be nonnegative rational numbers such that $\varepsilon \neq \delta$ and $\min\{6 - \max_j B_j, 9 - \max_{ij} X_{ij}\} > \epsilon$ and $\delta < 1$. Add the following intervals to the original graph:

| Interval | Occurrences | Clique ID |
|---|---|---|
| $(0, 1 + \varepsilon)$ | $t + 1$ times | I |
| $(1 + \varepsilon, 2 + \varepsilon)$ | $t + 1$ times | II |
| $(2 + \varepsilon, 6 - \varepsilon)$ | $t + 1$ times | III |
| $(6 - \varepsilon, 9 - \varepsilon)$ | $t + 1$ times | IV |
| $(9 - \varepsilon, 11 + \varepsilon)$ | $t + 1$ times | V |
| $(11 + \varepsilon, 13 - \varepsilon)$ | $t + 1$ times | VI |
| $(13 - \varepsilon, 14 - \varepsilon)$ | $t + 1$ times | VII |
| $(14 - \varepsilon, 14)$ | $t + 1$ times | VIII |
| $(0, 3 + \delta)$ | $\binom{2t}{2} - t^2 + t$ times | I |
| $(3 + \delta, 6 + \delta)$ | $\binom{2t}{2} - t^2 + t$ times | III |
| $(6 + \delta, 9 + \delta)$ | $\binom{2t}{2} - t^2 + t$ times | IV |
| $(9 + \delta, 14)$ | $\binom{2t}{2} - t^2 + t$ times | V |
| $(0, 14)$ | $\binom{2t}{3} - t^2$ times | I |

We will consider the optimal cost chromatic partition problem problem with 1 color of weight 0, $2t$ colors of weight 1, $\binom{2t}{2}$ colors of weight 2, $\binom{2t}{3}$ colors of weight 3, and $\binom{2t}{4}$ colors of weight 4. This is exactly the coloring version of the minimum cost intervention design problem.

We argue it is NP-complete to decide if the coloring cost is $3\binom{2t}{3} + 2\binom{2t}{2} + 14t^2 + 7t$. We reduce from numerical three dimensional matching. From the original reduction by [21], we see that if there is a solution to numerical three dimensional matching problem, then there exists a coloring of cost at most $3\binom{2t}{3} + 2\binom{2t}{2} + 14t^2 + 7t$.

Call the vertices with an $\varepsilon$ in their description the $\varepsilon$-class, and the intervals with a $\delta$ in their description the $\delta$-class. We see that the $\varepsilon$-class intervals can be partitioned into $t + 1$ contiguous regions, and the $\delta$-class can be partitioned into $\binom{2t}{2} - t^2 + t$ contiguous regions. By the choice of $\varepsilon$ and $\delta$, we also see that if the coloring does not follow this structure, then it takes more than $\binom{2t}{2} - t^2 + 2t + 1$ colors to color all these intervals. Further, there is a "gap" that cannot be filled by one of the original intervals. From the original hardness proof by [21], we see that if the coloring creates a gap in the original vertices that can be filled by a member of the $\varepsilon$-class or $\delta$-class, then it takes more than $2t^2$ colors to color the original intervals. We conclude that if the coloring of the $\varepsilon$-class and the $\delta$-class do not partition these intervals into contiguous regions, then the coloring must use a color of weight 4.

Again using the clique argument to show that the original problem is NP-complete, we see that if a coloring uses a color of weight 4, then the cost of this coloring is strictly more than $3\binom{2t}{3} + 2\binom{2t}{2} + 14t^2 + 7t$.

In the original reduction by [21], they prove that if there is a solution to the numerical three dimensional matching problem, the optimal coloring must have $t$ color classes of size 7, $t^2 - t$ color classes of size 5, and $t^2$ color classes of size 3. Introducing these new intervals, we see that the color classes of size 8 should take the weight 0 color and $t$ of the weight 1 colors, the $t$ color classes of size 7 should take the rest of the weight 1 colors, the $t^2 - t$ color classes of size 5 should take $t^2 - t$ colors of weight 2, the $\binom{2t}{2} - t^2 + t$ color classes must take the rest of the weight 2 colors, the $t^2$ color classes of size 3 should take $t^2$ weight 3 colors, and the $\binom{2t}{3} - t^2$ color classes of size 1 should take the rest of the weight 3 colors. By looking at the coloring of the original intervals, if the total cost is at most $3\binom{2t}{3} + 2\binom{2t}{2} + 14t^2 + 7t$, we can create a solution to the numerical three dimensional matching problem.

We can thus conclude that the unweighted minimum cost intervention design problem problem is NP-hard on interval graphs.