[Reviews · NeurIPS 2018]

Reviewer 1



The authors consider finding optimal set of intervention experiments to uniquely identify a causal graph from a DAG eq. class, where a cost can be given to intervention to each variable. They show that this problem is NP-hard. Since the basic formulation may produce large intervention sets, the authors also consider the case where each experiment has only a limited number of experiments. The authors develop approximations that have guarantees and perform well in simulations. The paper is well and clearly written. The referenced material covers the important literature to my knowledge. The NP-hardness result is interesting extension to existing similar results. The approximations are interesting as well, but not sure how practically relevant such results are (see point below). The setting in which we know the DAG eq. class is a bit restricting but still well motivated. The design problem refers to G that is a chordal graph, that is the result of a PC-algorithm run. The writers could point out how this relates to the general problem of having no information to start with, or having arbitrary information (not a DAG eq class, because we can go beyond that with ANM/LiNGAM or perhaps we have background knowledge or some experiments at hand already). The authors could/should still point out [20]:s thm 6 that shows that finding minimum size set of intervention experiments is NP-hard. This a very related result and discussion is in order (there seems to be also some discussion in p. 3063 of [11]). Simulations are sufficient, but not very extensive. In simulations, the authors compare to an IP solution, showing that their procedure is nearly optimal. But, how was this IP formulated? What was the running time? Does it make any sense to use the approximation in practical situations or is it merely a theoretic result? Fig 7 -> Fig 1. AFTER REBUTTAL: The IP solution time falls into the almost trivial -- with the current size networks the approximation does not seem to be practically relevant but only theoretical -- one can just use IP and wait. For different simulation setting the situation may be different. No update to scores.

Reviewer 2



The authors address the problem of designing minimum-cost interventions to fully learn a structural causal model. Specifically, the authors establish an equivalence between graph separating systems and graph coloring such that they can define an equivalent coloring version of the minimum cost intervention design problem. This allows them to specify an a (2 + epsilon)-approximation algorithm. In addition, they show that the minimum-cost intervention design problem is NP-hard, The results are useful and clearly presented. A somewhat unrealistic assumption of the paper is that intervention (randomization) is possible on any variable (and any set of variables simultaneously). In most real systems, the set of variables for which arbitrary interventions are possible is often very small. Medical researchers, for example, cannot realistically intervene on the gender, age, height, genetics, or occupation of a patient, to name just a few such variables. The paper would be improved by discussing scenarios in which the conditions assumed in the paper are realistic. A significant weakness of the paper is that the author assumes causal sufficiency (no latent confounders) despite the fact that experiments provide a powerful tool to avoid this assumption. Causal sufficiency is one of the least appealing aspects of the PC algorithm, and this is why substantial work has been done on algorithms that elimiante the causal sufficiency assumption (such as FCI and its variants). Another significant weakness is the assumption that the graph learned through observational data is correct, up to edge orientation. A large amount of empirical results show that this is unlikely in practice, and the proposed approach does not attempt to check whether the essential graph could be incorrect. The paper poorly defines interventions, experiments, or both. Section 1 states that “An intervention is a randomized experiment where we force a variable to take a particular value.” Experiments randomize some variables (omit all causes and generate the variable from a marginal distribution) and control others (hold their value constant). Interventions are usually characterized as the latter. The paper misstates the relationship between direct connection in the DAG and causation. For example, the authors state that “Given a causal graph, a variable is said to be caused by the set of parents in the graph.” In actuality, a variable is said to be caused by each of the set of ancestors.

Reviewer 3



Summary: The authors study the problem of designing experiments for learning causal structure under the assumption that each variable has a cost to intervene on, and the objective is to minimize the total experiment cost. It is first shown that the problem is in general NP hard, and then a greedy algorithm is proposed which is proven to be an approximate algorithm. Furthermore, the case that in each intervention, a limited number of variables can be intervened on is considered. An approximate algorithm is proposed for this case as well. Finally, the proposed approach is evaluated on synthetic data. The work for most of the part, is written clearly and is easy to understand. Comments: - In line 69, the vice versa part does not seem to be necessarily true. - In line 102, when we force a variable to just take a certain value, it may not necessarily change the distribution of its children. Also, what does the authors mean by "new skeleton" in line 105? - In Definition 2, the minimum cost intervention design problem is defined as finding a graph separating system of size at most m. What if there is no such separating system? I recommend changing it to minimum possible size, or m>m^*, where m^* is the minimum possible size. - In Definition 4, it seems that in reality we always need to have |I_i|=k. If so, can we say that Definition 5 is a special case of Definition 6? - How should we choose the cost values in practice? Do the authors recommend that these values should be defined manually by an expert? If so, this restricts the application of the proposed approach. - A major concern as usual regarding this work and similar work is its practicality aspects. It is not clear if the proposed method can be applied in practice, where there are latent confounders, statistical errors, undefined cost values, etc. - It seems that in reality the values of the costs may not follow a distribution similar to the design of experiments. I expect much larger costs following a heavy tailed distribution, or even more realistically, many variables should have cost equal to infinity, representing that intervention on those variables are not possible. Especially the latter situation is not considered in this work. - The cost quantization step seems unjustifiable in practice and seems to be only useful for theoretical guarantees to work. Clearly, this step can possibly change the significance of costs and make the distribution of costs close to uniform. - One important missing experiment in the last section is evaluating the approach on graphs with different sparsities. - Also, it would have been more useful if the authors performed simulations with the initial observational test step, so that we can see the effect of that step on the total performance as well. This will include error in estimating the skeleton, not having chordal graph, etc. - The proof of proposition 10 in the supplementary materials is not clear. Especially, it is not clear why S should be an independent set. This is important as Theorem 11 is also using this proposition. - One question regarding the presentation of the work: the case of restricting the number of intervened variables in each intervention is the general case, and the whole section 5 could be considered as a special case of that. The same is true for definition. I was wondering if there was a special reason that the authors did not merge these results and definitions?